# 3D-DIoU: 3D Distance Intersection over Union for Multi-Object Tracking in Point Cloud

**DOI:** 10.3390/s23073390

**Published:** 2023-03-23

**Authors:** Sazan Ali Kamal Mohammed, Mohd Zulhakimi Ab Razak, Abdul Hadi Abd Rahman

**Affiliations:** 1Institute of Microengineering and Nanoelectronics (IMEN), Universiti Kebangsaan Malaysia, Bangi 43600, Malaysia; 2Department of Automotive Technology, Erbil Technology College, Erbil Polytechnic University, Erbil 44001, Iraq; 3Center for Artificial Intelligence Technology, Universiti Kebangsaan Malaysia, Bangi 43600, Malaysia

**Keywords:** multi-object tracking, point cloud, 3D-DIoU, DIoU-NMS, multistage data association, tracklets, motion prediction

## Abstract

Multi-object tracking (MOT) is a prominent and important study in point cloud processing and computer vision. The main objective of MOT is to predict full tracklets of several objects in point cloud. Occlusion and similar objects are two common problems that reduce the algorithm’s performance throughout the tracking phase. The tracking performance of current MOT techniques, which adopt the ‘tracking-by-detection’ paradigm, is degrading, as evidenced by increasing numbers of identification (ID) switch and tracking drifts because it is difficult to perfectly predict the location of objects in complex scenes that are unable to track. Since the occluded object may have been visible in former frames, we manipulated the speed and location position of the object in the previous frames in order to guess where the occluded object might have been. In this paper, we employed a unique intersection over union (IoU) method in three-dimension (3D) planes, namely a distance IoU non-maximum suppression (DIoU-NMS) to accurately detect objects, and consequently we use 3D-DIoU for an object association process in order to increase tracking robustness and speed. By using a hybrid 3D DIoU-NMS and 3D-DIoU method, the tracking speed improved significantly. Experimental findings on the Waymo Open Dataset and nuScenes dataset, demonstrate that our multistage data association and tracking technique has clear benefits over previously developed algorithms in terms of tracking accuracy. In comparison with other 3D MOT tracking methods, our proposed approach demonstrates significant enhancement in tracking performances.

## 1. Introduction

An important challenge in computer vision study is multi-object tracking (MOT), which identifies and keeps track a unique identification (ID) for each object of interest in a point cloud series while predicting the locations of all objects. MOT has many important theoretical research implications and practical applications. Systems for visual security surveillance, vehicle visual navigation [1], augmented reality [2], human–computer interface, high senstitivity audio-visual (AV) [3] to name a few, all heavily rely on MOT systems with well-behaved performances. There are several difficulties that can deteriorate tracking performances in real-world applications. These difficulties include the way an object interacts, occlusion, and how close certain objects are related to one another. These difficulties lead to many unwanted detection mistakes and errors, including bounding box drift and ID changes, which cause tracking performance to degrade severely. As a result, this work proposes an improved and reliable MOT method for point cloud scenarios. Previously developed three dimension (3D) multiple object tracking (3D MOT) algorithms [4,5,6,7,8,9] adopt the tracking-by-detection pattern. Across frames, the tracklets depend directly on the 3D bounding boxes from 3D detectors.

In general, the concept of the tracking-by-detection algorithm consists of four modules: (i) input detection pre-processing module, (ii) motion module, (iii) association module, and (iv) managing tracklet life cycle. All objects of interest from the point cloud series are determined using the detector. Then, the identical objects from the detector and predicted motion model are associated using the metrics, which is established on features. A continually updating tracklet set is created by connecting the same item in many point cloud frames. In this procedure, the detector’s effectiveness and the performance of the data association algorithm jointly impact the tracking accuracy and flexibility. This detection process is normally evaluated using an intersection over union (IoU) metric.

The association results can be wrong when the input detectors are inaccurate. However, refining these detectors by using the non-maximum suppression (NMS) technique can improve the association. Additionally, we found that the association metric expressed between two 3D bounding boxes should be designed properly. Neither generalized IoU or GIoU [10] nor L2 [11] work well. The inference speed of the tracking system is significantly influenced by both the detector and the data association. Therefore, the multistage association process between predictions and tracklets can express the existence of the objects. Based on these findings, using distance-IoU (DIoU) over the tracking pipeline can significantly improve the solutions. In order to tackle 3D MOT issues, we propose an improved DIoU method in this paper. Consequently, we utilize Waymo Open Dataset [12] and nuScenes [13] in order to evaluate and verify our proposed algorithm. Our method and contributions, in brief, are as follows:We added DIoU-NMS to 3D MOT tracking pipeline and analyze the performance;We proposed the use of DIoU for two-stage and multi-stage data association, which showed competitive results on both Waymo Open Dataset and nuScenes;We used unmatched tracklets and unmatched detection from previous stages for data association in the next stage, and the verification results on Waymo Open Dataset show better performance for cyclist objects.

By using DIoU in the tracking pipeline, we overcome premature tracklet termination where the tracking framework depends on prediction position for invisible objects. Previous work in [14] used GIoU for the tracking process, which terminates an unassociated tracklet. Instead, we used DIoU to maintain the unassociated tracklet by using its predicted position. Therefore, when a temporarily invisible object reappears, it can be associated with its original predicted position.

## 2. Related Work

Point clouds are used in 3D multi-object tracking (MOT), which works in conjunction with the detection process of 3D objects in the autonomous driving challenge. The task of connecting objects together in a complete sequence is handled by 3D MOT, which is sensible of object location in all point cloud frames. In this process, temporal consistency is vital in addressing the tracking issue. The difference between a 3D MOT system and a 2D MOT system is that 3D MOT system uses a 3D space for the detecting procedure. Recent studies have been using 3D point cloud data for MOT applications, even without the use of extra features such as RGB data.

### 2.1. Two-Dimensional MOT Methods

Based on data association, recent 2D MOT systems can be categorized into batch and online techniques. The batch approach utilizes a full sequence search for a global optimum association. Meanwhile, authors in [15] proposed a TADT system in order to learn target-aware features for better recognition of the targets under variance appearance changes. While trackers use a maximum overlap technique based on IoU values to solve this concern, there are imperfections in the IoU values that make it impossible to continuously optimize the objective function when a provided bounding box is completely contained within or without another bounding box; this makes accurate estimation of the target state extremely difficult.

Meanwhile, authors in [16] designed a tracking method based on a distance-IoU (DIoU) loss for an estimation and classification of a target. Learning to track procedures used in many fields, a method in [17] employed MOT for guiding drones and controlling them. Correspondingly, an MOT is used in unmanned aerial vehicles (UAVs) for collision prevention in [18]. Detecting lanes in driver assistance systems is challenging under bad climatic changes, hence a method [19] introduced a two-tier deep learning-based lane detection system for many images at a different number of weather situations. Texture features are extracted and an optimized deep convolution network is used for road and lane classification. On the other hand, authors in [20] proposed tracking algorithms that measure the dynamic similarity of tracklets and recover missing data due to long occlusions due to motion dynamics that provide strong cues while tracking targets with identical or very similar appearance. However, the algorithms are limited to 2D objects. 

However, detections from the previous frame and the current frame are accessible to the tracker, the SORT method [21] proposed online tracking-by-detection, the tracker component runs at 260 Hz for updating its states which is useful for real-time applications. In contrast, our work uses 3D point cloud data, and the system works near real-time with 10 Hz on Waymo Open Dataset and 2Hz on nuScenes.

Several online 2D trackers [22,23,24,25] have suggested improved detection qualities and utilize the tracking-by-detection paradigm. Unfortunately, due to scale variance, the object items in RGB pictures vary in size, making association and motion models more difficult. However, 2D MOT may readily make use of rich RGB information and employ appearance models based on online learning process [24,26,27,28]. The design of MOT frameworks should be compatible with data taken from LiDAR or a camera.

### 2.2. Three-Dimensional MOT Methods

Previous research in 3D multi-object tracking that followed the tracking-by-detection paradigm often solved tracking problems by using a bipartite graph-matching mechanism on top of ordinary detectors. Depending on early works on 2D MOT [13,14,15], many strategies emphasize enhancing the relationship between detection and tracklets by simulating their movements or attendance, or a mix of the two. A state of Kalman filter in [29] is specified on the 2D plane. An AB3DMOT method [7] offers a baseline technique built on the PointRCNN detector [30] that combines the 3D Kalman filter with the Hungarian algorithm [31]. While AB3DMOT utilized 3D IoU for the association process, Chiu et al [32] employed Mahalabnobis distance [10] as an alternative. On the other hand, GIoU [33] is used in Simple Track [14] for the 3D association. The 2D velocity of the detected box is predicted by learning in CenterPoint [8] followed by CenterTrack [34] and performed simple point–distance corresponding. In addition, an aspect of labeling for 3D objects in self-driving vehicles are discussed in [35]. 

To further prevent misperception throughout the association procedure, GNN3DMOT [36] used a Graph Neural Network to collect appearance and motion data in order to establish feature interaction between objects. Authors in [37] proposed a probabilistic multi-modal structure that covered trainable sections for 2D and 3D object feature fusion, distance space arrangement, and trajectory creation. A method in [37] joined 2D and 3D object indication gained from 2D and 3D detectors. The authors in a single graph form combined the prediction models with object identification characteristics [9]. In this paper, on the other hand, we used a simple DIoU metric for data association.

Behind early knowledge in 2D MOT [38,39,40], the preceding works in 3D MOT [5,7,8,9,24,26] frequently implemented a counting-based method for tracklet life cycle management. Different tracklets are created for each frame with detected objects that are not related to any current tracklet. The tracklets that lose their targets for a number of frames (usually fewer than 5) are terminated. Authors in [4,27] recommended that tracks are initiated and terminated based on their confidence score value, which is calculated from the confidence measurement of their related detections. Nevertheless, predictors that are not related with fresh detections are permanently terminated. In contrast, we show that by positively anticipating and conserving the object-predicted box, predictors that have lost their targets may be appropriately preserved for future association.

## 3. Materials and Methods

In this section, a simple tracking with the PointPillars and a motion prediction technique is proposed, and the workflow of tracking procedure is shown in Figure 1. The tracking process consists of the following parts:Detection: for this step, the bounding boxes are selected from the detector, as shown in Figure 2;Selected Detection: by applying NMS process the number of bounding boxes decreased and the unwanted boxes are removed;Tracklets, Prediction, and Motion Update: all these processes are related to each other where Kalman filter is used, as illustrated in Figure 3 and Figure 4;Multi-Stage Association: in this step the detectors in the present frame are associated with the tracklets from the previous frame. The unmatched prediction and tracklets are associated in another stage. Three-dimensional GIoU and DIoU association metrics are used in this work is coupled with Hungarian algorithm;Motion update and Life Cycle Management: the creation and termination of the tracklets are updated and are determined in this step, and the final tracklets are shown in Figure 5.

### 3.1. Adding 3D DIoU Function

In this part, we examine and enhance the detection and multi-stage association modules by including a 3D DIoU model. In this work, we revised the NMS and its association function for bounding boxes of the conventional tracking method to enhance the tracking capability of multiscale and occluded objects.

The association speed and performance of the object tracker are directly dependent on the values of the association function. To determine the multiple object tracking (MOT) value, it is necessary to calculate the correspondence among the bounding boxes using the tracking method. In order to determine the volume of union between two bounding boxes, the intersection over union (IoU) metric [41] are used, and the consistent association function is stated as follows:(1)$$IoU=\frac{\left|B_{1}\cap B_{2}\right|}{|B_{1}\cup B_{2|}},$$
where $B_{1}$ and $B_{2}$ are 3D bounding boxes; $\left|B_{1}\cap B_{2}\right|$ indicates the volume of intersection of $B_{1\;}$ and $B_{2}$; and $\left|B_{1}\cup B_{2}\right|$ indicates the volume of the union of $B_{1}$ and $B_{2}$. The IoU is equal to 0 when there is no intersection between the two 3D bounding boxes. In this case, the tracking process cannot continue.

As a solution for this gradient vanishing matter, generalized intersection over union (GIoU) equation [33] is used in the tracking technique, which is stated as follows:(2)$$GIoU=IoU-\frac{\left|D\right|}{\left|C\right|},$$
where C  is the smallest volume that covers $B_{1}$ and $B_{2}$; let $D=C/\left(B_{1}\cup B_{2}\right)$, the $C=B_{1}\cup B_{2}\cup D;\left|C\right|$ and $\left|D\right|$ stand for the volume of C and D, respectively. When $B_{2}$ box contains the $B_{1}$ box, then the variance between each $B_{1}$ box and the $B_{2}$ box are the same, GIoU, in this case, degenerates into IoU, without any tracking relationship.

IoU and GIoU only take into account the overlapping volume, and the associated functions have two drawbacks, including delayed corresponding and incorrect association. However, distance intersection over union (DIoU) uses the standardized distance among the centers of the $B_{1}$ and $B_{2}$ bounding boxes. The following definition relates to this association function [42]:(3)$$DIoU=IoU-\frac{d^{2}}{c^{2}},$$
where d  is the Euclidean distance length between the center points of the $B_{1}$ and $B_{2}$ bounding boxes; c  is the diagonal length of the smallest enclosing box that encompasses the two boxes. The DIoU function causes the model to acquire quick association if the two boxes are in either the horizontal or vertical direction at the same time. Directly reducing the normalized distances between central point’s using the DIoU function leads to a faster convergence rate [42] and more precise association. The IoU, GIoU, and DIoU, as expressed above, are used to describe the association between any two bounding boxes. The algorithm of 3D DIoU metric is defined as Algorithm 1.


**Algorithm 1. 3D Distance Intersection Over Union Function**
Input: the information data of $B_{1}$ and $B_{2}$ bounding boxes:$B_{1}=\left(x_{1},\;y_{1},z_{1},\;l_{1},\;w_{1},h_{1},\theta_{1}\right)$$,\;B_{2}=\left(x_{2},\;y_{2},\;z_{2},\;l_{2},\;w_{2},\;h_{2},\;\theta_{2}\right)$
Output: 3D DIoU Association Metric1. Determining the Projections ${B^{\prime }}_{1}$ and $B_{2}^{\prime }$ of $B_{1}$ and $B_{2}$on the bird’s eye view, respectively $B_{1}^{\prime }=\left(x^{1}{}_{1},\;y^{1}{}_{1},\;x^{2}{}_{1},\;y^{2}{}_{1},\;{\theta^{\prime }}_{1}\right)$, $B_{2}^{\prime }=(x_{2}^{1},\;y_{2}^{1}$$,\;x_{2}^{2},\;y_{2}^{2},\;\theta_{2}^{\prime }$)2. 
$A_{1}\leftarrow$the area of the 2D box
$B_{1}^{\prime }$
3. 
$A_{2}\leftarrow$the area of the 2D box
$B_{2}^{\prime }$
4. 
$I_{2D}\leftarrow$intersection between
$B_{1}^{\prime }$ and
$B_{2}^{\prime }$
5. 
$U_{2D}\leftarrow$union between
$B_{1}^{\prime }$ and
$B_{2}^{\prime }$
6. 
$I_{h}\leftarrow$the height of the intersection between
$B_{1}$ and
$B_{2}$
7. 
$U_{h}\leftarrow$the height of the union between
$B_{1}$ and
$B_{2}$
8. 
$I_{w}\leftarrow$the width of the intersection between
$B_{1}$ and
$B_{2}$
9. 
$U_{w}\leftarrow$the width of the union between
$B_{1}$ and
$B_{2}$
10. 
$I_{l}\leftarrow$the length of the intersection between
$B_{1}$ and
$B_{2}$
11. 
$U_{l}\leftarrow$the length of the union between
$B_{1}$ and
$B_{2}$
12. 
$I_{v}\leftarrow$the volume of the intersection between
$B_{1}$ and
$B_{2}$
13. 
$U_{v}\leftarrow$the volume of the union between
$B_{1}$ and
$B_{2}$
14. $d_{x}\leftarrow$the center distance for $(x_{1}-x_{2}$)15. $d_{y}\leftarrow$the center distance for $(y_{1}-y_{2}$)16. $d_{z}\leftarrow$the center distance for $(z_{1}-z_{2}$)17. 
$d^{2}\leftarrow$the diagonal distance between
$B_{1}$ and
$B_{2}$
18. 
$c^{2}\leftarrow$the diagonal distance for the smallest enclosing box that encompasses between
$B_{1}$ and
$B_{2}$
19. If $I_{2D}\leq \;0:$   $I_{v}\;=\;0;$  else:  If $\;I_{h}\leq \;0:$   $I_{v}\;=\;0;$      else:        $I_{v}=I_{2D}\;X\;I_{h};$20. 
$d^{2}=d_{x}^{2}$
$+d_{y}^{2}$
$+d_{z}^{2};$
21. 
$c^{2}=U_{w}^{2}$
$+U_{I}^{2}$
$+U_{h}^{2};$
22. 
$IoU_{3D}=\frac{I_{v}}{U_{v}};$
23. 
$DIoU_{3D}=IoU_{3D}-\frac{d^{2}}{c^{2}}$


### 3.2. Non-Maxamium Suppressing (NMS) Upgrade to DIoU-NMS

To locate local maximum and to eliminate non-maximum bounding boxes, the NMS approach is used. Most object-tracking systems use NMS as a pre-processing stage, which is often used to choose the bounding boxes before starting the tracking operation. Depending on the score for classification confidence, which is the foundation of the original NMS, the bounding box that has the highest confidence score can be maintained. Since IoU and classification confidence scores are typically not strongly correlated, it is difficult to pinpoint many classification labels with a high number of confidence scores. When using the tracking method with the original NMS technique, analysis is only performed over overlapping regions, increasing the likelihood of missing and false detection, specifically in scenes with extremely overlapping objects.

We use DIoU-NMS to increase the detection efficiency for the occluded object. DIoU-NMS uses DIoU as a tool for suppressing the redundant bounding boxes, in contrast to the original NMS, which uses IoU as the criterion. DIoU-NMS takes into account the distance length between the center points of the two bounding boxes in addition to the overlapping area. The DIoU-NMS is stated as
(4)$$s_{i}=\left\{\begin{array}{c} s_{i},\;\;\;IoU-R_{DIoU}\left(M,B_{i}\right)<\varepsilon \\ 0,\;\;\;\;IoU-R_{DIoU}\left(M,B_{i}\right)\geq \varepsilon \end{array}\right.,$$
where $s_{i}$ stands for the score of classification confidence; IoU is stated in an Equation (1); ε denotes the value of NMS threshold; M represents the highest-scoring bounding boxes; and $B_{i}$ is the pending bounding box. When conducting DIoU-NMS, the distance between the centers of two bounding boxes is taken into account concurrently with IoU. The distance is indicated by $R_{DIoU}$ and the equivalent equation is as follows:(5)$$R_{DIoU}=\frac{p^{2}\left(b_{1},b_{2}\right)}{c^{2}}$$
where $p^{2}$ denotes the length of the central distance measured between the bounding box b center point and the bounding box $b_{2\;}$ ones; $c^{2}$ is the smallest box’s diagonal, which contains both boxes.

## 4. Results

### 4.1. Datasets

There have been several MOT datasets recommended and used during the last few years. Waymo Open Dataset [12] and nuScenes [13] are the most commonly used and most considerable benchmark for MOT. Waymo Open Dataset (WOD) includes a perception dataset and a motion dataset. The total number of scenes in the dataset is 1150, divided into 150, 202, and 798 scenes for testing, validation, and training, respectively. While the motion dataset comprises 103,354 sequences, the perception dataset has 1950 lidar sequences that have been annotated. Each sequence is recorded for 20 s at a sample rate of 10 Hz. For each frame, point cloud data and 3D ground truth boxes for vehicles, pedestrians, and cyclists are provided. By using the evaluation metrics stated in [12], we recorded multiple object tracking accuracy (MOTA), multiple object tracking precision (MOTP) [43], Miss, Mishmatch, and false Positive (FP) for objects with the L2 diffuclty level.

NuScenes [13] provides ground truth 3D box annotations at 20 frames per second and LiDAR scans at 2 frames per second (fps) for a total of 1000 driving sessions. We report identity switches (IDS), AMOTA [7], and MOTA for nuScenes. AMOTA, the average value of MOTA, serves as the main indicator for assessing 3DMOT on nuScenes, is created by merging MOTA over several recalls. Meanwhile, AMOTP, the average value of MOTP, indicates an error value for the association process. Hence, the value for MOTP and AMOTP should be kept as small as possible.

### 4.2. DIoU-NMS Results

Our approach aims to increase the precision without considerably reducing the recall. We apply a strict DIoU-NMS to the input detections, and it is found that the ID switch only recorded 479 switches, in comparison with the IoU method, which is 519, as shown in Table 1.

In addition, when DIoU-NMS is applied to the Waymo Open Dataset, the MOTA is higher than that resulting from IoU-NMS. Similarly, the mismatch value improved and reached 0.077% for vehicle class, as in Table 2. Meanwhile, MOTA results reached 51% and the mismatch value is equal to 0.4% for pedestrian objects, as shown in Table 3. 

### 4.3. Association Results

We used 3D box detection from the CenterPoint method as the input data. To select boxes with scores higher than 0.7 on the Waymo Open Dataset, 3D IoU-NMS was set to 0.7. To associate between detection and prediction boxes, we used two-stage data association, namely 3D GIoU and DIoU. In this case, we associated the detection and prediction boxes by using DIoU in the first stage, then we re-associated again any un-associated boxes with DIoU for the detection and tracklets in the second stage. A similar second stage approach was applied to the third and higher order stage data association. The results for two-stage data associations are shown in Table 4 for vehicle class, and Table 5 for pedestrian class using the Waymo Open Dataset. The first row in both Table 4 and Table 5 represents two-stage data association results, where 3D GIoU metric is coupled with the Hungarian algorithm to match between detections and tracklets. On the other hand, in the second row, we used 3D DIoU instead of GIoU metrics.

On the other hand, Table 6 represents a three-stage data association, where only the cyclist class has a low false positive value (FP), the unmatched detections and unmatched tracklets are associated using 3D DIoU coupled with the Hungarian algorithm.

### 4.4. Comparison with Previous Techniques

In this part, we incorporate the aforementioned methods into the combined DIoU-NMS and DIoU data association in order to demonstrate how the performance can be enhanced. Table 7 below shows our proposed 3D MOT trackers perform better than the baselines. In the case of the Waymo Open Dataset, although the dimensions of vehicles and pedestrians are much different, DIoU-NMS and two-stage DIoU data association are adequate and appropriate for both vehicle and pedestrian objects due to high tracking performance values, as the results are illustrated in Table 7 and Table 8. The comparison for vehicle class in the Waymo Open Dataset test set is tabulated in Table 7, which CenterPoint [8] recognitions are utilized. For comparison, the results from AB3DMOT [7] and Chiu et al. [5] are also presented. On the same note, Table 8 highlights the results for pedestrian class in the Waymo Open Dataset test set. Meanwhile, the three-stage technique is only applicable to cyclist objects due to the limitation of MOTA value computation for vehicles and pedestrians in this multi-stage evaluation. DIoU-NMS shows effective results on the nuScenses dataset, as shown in Table 9. In this case, CenterPoint [8] detection is utilized and compared. In all tests, 2 Hz frame rate is used for the detection.

### 4.5. Comparison between GIoU and DIoU

As a comparison between the association metric between GIoU and DIoU, the score threshold for selecting the boxes is set equal to 0.7. The GIoU association threshold is equal to 1.5 and in the case of DIoU, it is equal to 1. Meanwhile, the NMS-IoU threshold is equal to 0. The figures below show the associations between the detection results (green boxes) and the predicted results (blue boxes). In this case, it can be seen that for both DIoU and GIoU in the first stage, and for DIoU, the predicted box is preserved until the object is detected again. Meanwhile, in the GIoU case, the predicted box is terminated when the object is temporarily not observed, causing an identity switch, as illustrated in Figure 6. At frame 9, which the figures shown in the first row, the detected box for vehicle ID number 2 (green box) is associated with its predicted box (blue box). The second row showed frame 11, which contains the tracklet for vehicle ID 10. We also use DIoU with a predicted box for vehicle ID 2 for association process. However, when we apply GIoU on the association process, we obtained the tracklet only for vehicle ID number 10 and the predicted box for vehicle ID 2 is terminated, as shown by the first column in the second row of Figure 6. At frame 28, the tracklet for vehicle 3 and 11 and predicted position for vehicles 0, 1, 2, and 6 are shown in final row, second column when we apply DIoU on the association process. On the other hand, when we apply GIoU on association process, we obtain the tracklet for vehicle 3 and 11 only, and the predicted boxes are terminated for vehicles 0, 1, 2, and 6, which lead to an increase in ID switch and lowering of the multiple object tracking accuracy (MOTA) value.

## 5. Conclusions

It was discovered that tracklet termination leads to identity switches in 3D MOT, which are common and unresolved issues in recent 3D MOT studies. Therefore, in this paper, we proposed a hybrid method of using DIoU-NMS and DIoU in order to improve the association between tracklet and prediction boxes for objects. We found that by using the combination of DIoU-NMS and DIoU, the identity switch cases can be reduced.

Additionally, we used DIoU for multi-stage association, which lead to an increase in MOTA values for small objects on the Waymo Open Dataset. Experiment results show that DIoU-NMS can significantly reduce the identity switches when it is used in selecting the detectors for tracking. Our approach achieved 479 ID switches for the vehicle objects on nuScense compared with 519 for GIoU only. While the mismatches were improved slightly for vehicles and pedestrian objects, the MOTA results also recorded better performance in tracking on the Waymo Open Dataset. Meanwhile, two-stage data association results demonstrated significant improvements in MOTA values with 58.9% and 59.7% for vehicle and pedestrian objects, respectively. The FP values also significantly improved, which are 11.1% and 47.7% for vehicle and pedestrian objects, respectively. In addition, using DIoU for three-stage association reduced the false positive detection as well as improved MOTA values.

In comparison to previous work, our method recorded significant improvement in ID mismatches, which achieved at least 63.8% and 28.6% reductions for vehicle and pedestrian objects, respectively. Similarly, test results on Waymo Open Dataset show MOTA values for both vehicles and pedestrian objects reach over 60%, overtaking all previously issued LiDAR-based methods. The results show great potential for future 3D MOT analysis and can pave the path for many real-time 3D tracking-by-detection applications.

## Figures and Tables

**Figure 1 sensors-23-03390-f001:**
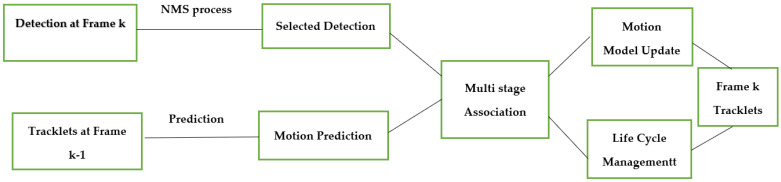
Three-dimensional MOT workflow steps.

**Figure 2 sensors-23-03390-f002:**
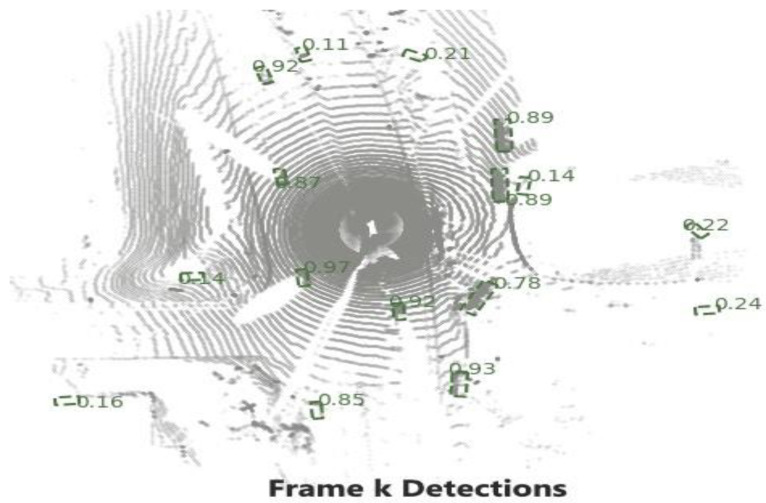
Vehicle bounding box detectors.

**Figure 3 sensors-23-03390-f003:**
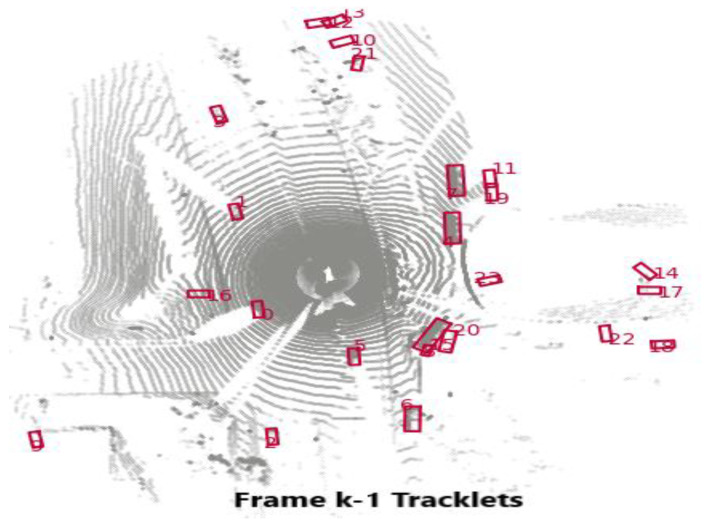
Vehicle tracklets bounding boxes.

**Figure 4 sensors-23-03390-f004:**
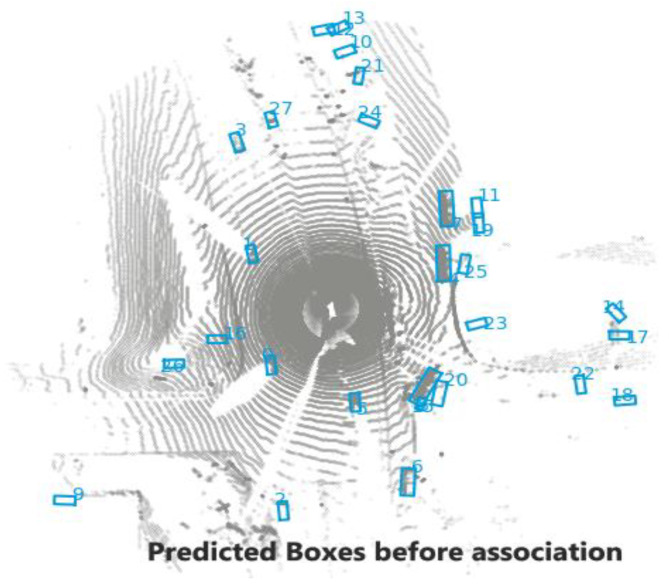
Motion prediction process.

**Figure 5 sensors-23-03390-f005:**
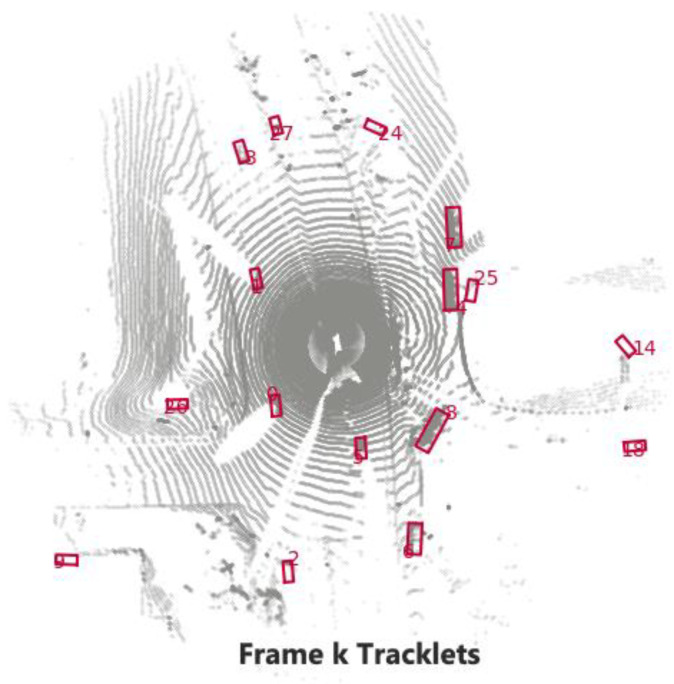
Tracklets at frame k.

**Figure 6 sensors-23-03390-f006:**
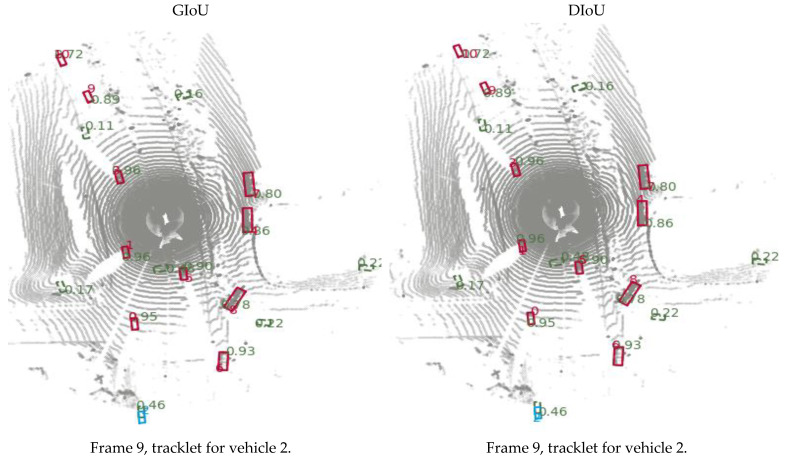
Comparison between GIoU and DIoU for Association process.

**Table 1 sensors-23-03390-t001:** NMS for IoU and DIoU in the detection process with GIoU in association two stage.

NMS Metric	AMOTA	AMOTP	RECALL	MOTA	ID Switch
IoU	0.687	0.573	0.725	0.592	519
DIoU	0.688	0.573	0.722	0.592	479

**Table 2 sensors-23-03390-t002:** Comparison of the tracking results for vehicle objects using different NMS metrics on the validation set of Waymo Open Dataset.

NMS Metric	MOTA	MOTP	Miss	Mismatch (%)	FP
IoU	0.544	0.168	0.355	0.08	0.099
DIoU	0.547	0.1678	0.354	0.077	0.097

**Table 3 sensors-23-03390-t003:** Comparison of the tracking results for pedestrian objects using different NMS metrics on the validation set of Waymo Open Dataset.

NMS Metric	MOTA	MOTP	Miss	Mismatch (%)	FP
IoU	0.505	0.311	0.397	0.45	0.092
DIoU	0.510	0.311	0.388	0.40	0.097

**Table 4 sensors-23-03390-t004:** Comparisons for 3D MOT two-stage association on vehicle class, Waymo Open Dataset validation set.

Two-Stage	MOTA	MOTP	Miss	Mismatch (%)	FP
GIoU	0.5612	0.1681	0.3344	0.078	0.1035
DIoU	0.5892	0.1736	0.3165	0.1559	0.092

**Table 5 sensors-23-03390-t005:** Comparisons for 3D MOT two-stage association on pedestrian class, Waymo Open Dataset validation set.

Two-Stage	MOTA	MOTP	Miss	Mismatch (%)	FP
GIoU	0.5776	0.3125	0.3090	0.425	0.1091
DIoU	0.5972	0.3518	0.3360	0.96	0.0570

**Table 6 sensors-23-03390-t006:** Three-dimensional MOT three-stage association on cyclist class, Waymo Open Dataset validation set.

Three-Stage	MOTA	MOTP	Miss	Mismatch (%)	FP
DIoU	0.6018	0.2855	0.3033	0.6613	0.0881

**Table 7 sensors-23-03390-t007:** Comparison on Waymo Open Dataset test set, vehicle class.

Method	MOTA	MOTP	Mismatch (%)
AB3DMOT [7]	0.5773	0.1614	0.26
Chiu et al. [5]	0.4932	0.1689	0.62
CenterPoint [8]	0.5938	0.1637	0.32
This Work	0.6061	0.1738	0.094

**Table 8 sensors-23-03390-t008:** Comparison on Waymo Open Dataset test set, pedestrian class.

Method	MOTA	MOTP	Mismatch (%)
AB3DMOT [7]	0.5380	0.3163	0.73
Chiu et al. [5]	0.4438	0.3227	1.83
CenterPoint [8]	0.5664	0.3116	1.07
This Work	0.615	0.329	0.521

**Table 9 sensors-23-03390-t009:** Comparison on the nuScenes test set.

Method	AMOTA	AMOTP	MOTA	ID Switch
AB3DMOT [7]	0.151	1.501	0.154	8987
Chiu et al. [5]	0.550	0.798	0.459	736
CenterPoint [8]	0.638	0.555	0.537	730
CBMOT [4]	0.649	0.592	0.545	517
OGR3MOT [9]	0.656	0.620	0.554	248
This Work	0.658	0.568	0.557	569

## Data Availability

Data supporting the conclusions of this manuscript are provided within the article and will be available from the corresponding author upon request.

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
