# Peer review of "3D-DIoU: 3D Distance Intersection over Union for Multi-Object Tracking in Point Cloud"

_sensors, 2023, doi:10.3390/s23073390_

Round 1

Reviewer 1 Report

The idea of the proposed work is not new but interesting. However, the following remarks should be considered in the revised version of the manuscript:

1. The introduction part is well written. However, a little extension is required to support the idea of 3D implementation over the limited performance of 2D alone.

2. The literature review part is weak. The authors are advised to add more and refer to the latest number of articles. Similarly, the optimization-based study should also be considered as it may enhance the capability of the proposed work. The following studies are suggested in the autonomous driving scenario:

a) Dewangan, D.K., Sahu, S.P. Lane detection in intelligent vehicle system using optimal 2- tier deep convolutional neural network. Multimed Tools Appl 82, 7293–7317 (2023). https://doi.org/10.1007/s11042-022-13425-7

3. The equation(s) must be numbered. However, there is no strong relevancy of mentioning equation(s) in the context, as they are generalized ones.

4. Comparison with the state-of-the-art should be improved by adding a few more studies.

5. Have the authors computed the time computation for the proposed approach. Consider this parameter and compare the results with other methods.

6. The discussion part is also weak. Discuss how the proposed approach solves the issues considering the other real-time situations.

Reviewer 2 Report

The paper presents a matching function for Multi-Object Tracking in point cloud. The association of tracklets depends on the accuracy of the input detector as well as the data association strategy. The proposal goes into achieving a higher performance by employing a matching function that employs Intersection over Union (IoU) measurement as well as a ratio between the Euclidean distance between boxes and the diagonal length of the smallest enclosing box. The Non-Maximum Suppressing algorithm with the proposed similarity measures is used to estimate the association between tracklets in successive frames.

However, the similarity measure employs the intersection over union (IoU) between the detected box and the ground truth box. This measurement only works if there is a ground truth for comparison. In a real-time application, no ground truth is provided. The similarity function should not include previously acquired information. The authors should explain how they propose to overcome this problem.

On page 4, the authors refer to a Figure 5 (line 63) that is not contained in the document. The reference should be corrected, or Figure 5 should be included.

The achieved performance barely outperforms using only IoU with a minor penalty in execution time. Furthermore, the proposed matching function outperforms GIoU while saving execution time. Therefore, the benefits of their proposal do not seem important. The authors should provide a discussion to establish the benefits of their proposal over classical problems of IoU, such as no intersection between boxes, to better showcase the benefits of their proposal.

There are several typos throughout the paper, from the title to the sections, section titles, and figures, which should be corrected. The writing style also requires improvement.

Round 2

Reviewer 2 Report

The title still displaying a typo: “Mutil-Object” extensive review should be performed to ensure that all grammar and spelling errors are corrected.

The wording of the benefits of using DIoU instead of GIOU on page 2 should be improved. The discussion is succinct.

Section 2.1. needs to be rewritten, there are several errors and it’s hard to follow.

Author Response

Manuscript ID: sensors-2149093

Dear Editor,

Recently your reviewers provided much-appreciated feedback on our submitted manuscript entitled “3D-DIoU:3D Distance Intersection over Union for Multi-Object Tracking in Point Cloud”. We hereby present our response to your reviewers' remarks, in which we provide justifications and describe the manuscript updates introduced in acknowledgment of their considerations. The revised manuscript with track changes was uploaded online.

Herewith, we list and answer all the reviewers’ comments as well as highlight all the changes in the manuscript. We hope these improvements to the original paper will add more clarity to our work, and thus establish the updated paper that will be accepted for publication in your journal.

Reviewer 1

No

Reviewer’s Comments

Answer

1

The introduction part is well written. However, a little extension is required to support the idea of 3D implementation over the limited performance of 2D alone.

We highlight our contribution results in line 68-83 page 2.

2

The literature review part is weak. The authors are advised to add more and refer to the latest number of articles. Similarly, the optimization-based study should also be considered as it may enhance the capability of the proposed work. The following studies are suggested in the autonomous driving scenario:

a) Dewangan, D.K., Sahu, S.P. Lane detection in intelligent vehicle system using optimal 2- tier deep convolutional neural network. Multimed Tools Appl 82, 7293–7317 (2023). https://doi.org/10.1007/s11042-022-13425-7

We make complete changes in the literature review part. See lines 94 page 2 to line 153 page 4.

3

The equation(s) must be numbered. However, there is no strong relevancy of mentioning equation(s) in the context, as they are generalized ones.

We add numbers to the equations

4

Comparison with the state-of-the-art should be improved by adding a few more studies.

We add more comparison studies of 3D MOT techniques

5

Have the authors computed the time computation for the proposed approach. Consider this parameter and compare the results with other methods.

We include a table of comparison between the GIoU and DIoU tracking performance, as in Figure 6.

6

 The discussion part is also weak. Discuss how the proposed approach solves the issues considering the other real-time situations.

In the literature review, we highlight about old method of simple, online real-time tracking by using a camera and the paper method by using lidar. See line 117-121 page 3. In the result section, we add a discussion on how our proposed method can potentially solve the real-time issue. (this is the part where we should add a little bit in the result section after Figure 6)

Reviewer 2 (First Version)

No

Reviewer’s Comments

Answer

1

The paper presents a matching function for Multi-Object Tracking in point cloud. The association of tracklets depends on the accuracy of the input detector as well as the data association strategy. The proposal goes into achieving a higher performance by employing a matching function that employs Intersection over Union (IoU) measurement as well as a ratio between the Euclidean distance between boxes and the diagonal length of the smallest enclosing box. The Non-Maximum Suppressing algorithm with the proposed similarity measures is used to estimate the association between tracklets in successive frames.

However, the similarity measure employs the intersection over union (IoU) between the detected box and the ground truth box. This measurement only works if there is a ground truth for comparison. In a real-time application, no ground truth is provided. The similarity function should not include previously acquired information. The authors should explain how they propose to overcome this problem.

In our configuration file, we choose between GIoU and DIoU, we use IoU only inside the function calculation and we correct the writing text in line 304-313 page 9.

2

On page 4, the authors refer to a Figure 5 (line 63) that is not contained in the document. The reference should be corrected, or Figure 5 should be included.

We correct the text in line 168 page 4

3

The achieved performance barely outperforms using only IoU with a minor penalty in execution time. Furthermore, the proposed matching function outperforms GIoU while saving execution time. Therefore, the benefits of their proposal do not seem important. The authors should provide a discussion to establish the benefits of their proposal over classical problems of IoU, such as no intersection between boxes, to better showcase the benefits of their proposal.

We highlight our contribution results in line 68-83 page 2. The benefit of using DIoU against GIoU in Table 10

4

There are several typos throughout the paper, from the title to the sections, section titles, and figures, which should be corrected. The writing style also requires improvement.

We re-wrote many sections and corrected the grammatical mistakes in the manuscript.

Reviewer 2 (Second Version)

No

Reviewer’s Comments

Answer

1

The title still displaying a typo: “Mutil-Object” extensive review should be performed to ensure that all grammar and spelling errors are corrected.

We correct the typo and have carefully  checked the entire manuscript

2

The wording of the benefits of using DIoU instead of GIOU on page 2 should be improved. The discussion is succinct.

We revised Section 1 as well as a paragraph of the benefits of using DIoU over GIoU in page 2, at the end of Section 1.

3

Section 2.1. needs to be rewritten, there are several errors and it’s hard to follow.

Section 2.1 has been revised. We also revise section 2.2

4

We add a new section 4.5 to compare GIoU and DIoU results

Kind regards,

Zulhakimi
